# Accurate Measurement of Temperatures in Industrial Grinding Operations with Steep Gradients

**DOI:** 10.3390/s24061741

**Published:** 2024-03-07

**Authors:** Iñigo Pombo, José Antonio Sánchez, Einar Martin, Leire Godino, Jorge Álvarez

**Affiliations:** 1Department of Mechanical Engineering, Bilbao School of Engineering, University of the Basque Country (UPV/EHU), 48013 Bilbao, Spain; joseantonio.sanchez@ehu.eus (J.A.S.); einar.martin@ehu.eus (E.M.); leire.godino@ehu.eus (L.G.); 2IDEKO S. Coop., 20870 Elgoibar, Spain; jalvarez@ideko.es

**Keywords:** grinding, temperatures, thermocouples, grinding wheels, thermal inertia

## Abstract

Due to the continuously growing demands from high-added-value sectors such as aerospace, e-mobility or biomedical bound-abrasive technologies are the key to achieving extreme requirements. During grinding, energy is rapidly dissipated as heat, generating thermal fields on the ground part which are characterized by high temperatures and very steep gradients. The consequences on the ground part are broadly known as grinding burn. Therefore, the measurement of workpiece temperature during grinding has become a critical issue. Many techniques have been used for temperature measurement in grinding, amongst which, the so-called grindable thermocouples exhibit great potential and have been successfully used in creep-feed grinding operations, in which table speed is low, and therefore, temperature gradients are not very steep. However, in conventional grinding operations with faster table speeds, as most industrial operations are, the delay in the response of the thermocouple results in large errors in the maximum measured value. In this paper, the need for accurate calibration of the response of grindable thermocouples is studied as a prior step for signal integration to correct thermal inertia. The results show that, if the raw signal is directly used from the thermocouples, the deviation in the maximum temperature with respect to the theoretical model is over 200 K. After integration using the calibration constants obtained for the ground junction, the error can be reduced to 93 K even for feed speeds as high as 40 m/min and below 20 K for lower feed speeds. The main conclusion is that, following the proposed procedure, maximum grinding temperatures can be effectively measured using grindable thermocouples even at high values of table speed.

## 1. Introduction

Grinding processes play a critical role in the manufacturing of high-tech components [1]. Due to the continuously growing demands from high-added-value sectors such as aerospace, e-mobility or biomedical (amongst others) bound-abrasive technologies are the key to achieving extreme requirements with regard to surface finish, fatigue life, tolerances, etc. The technological pull from industry is followed by scientific approaches to new problems from academia. Extensive research focused on the new technology needs has been published in recent years. For instance, gear grinding places very high demands that must be satisfied. Guerrini et al. [2] analyzed surface integrity in dry generation for the automotive sector. In [3], the process of gear profile grinding is addressed, analyzing the mechanisms of surface generation and surface roughness distribution. The manufacturing of aerospace components commonly involves advanced grinding techniques for low-machinability alloys, for which creep-feed grinding stands as an efficient solution [4,5]. The effect of the grinding process on the fatigue life of critical components for vacuum pumps has been investigated in [6]. Many other examples of the importance of grinding technologies can be found in the literature.

Because of the nature of the removal process during grinding, the values of specific energy are considerably higher than those of machining processes such as turning or milling. Generated energy is rapidly dissipated as heat [7], generating thermal fields on the ground part. The pattern of these thermal fields is characterized by high temperatures and very steep gradients, localized on the area of contact between the wheel and workpiece. The consequences on the ground part are broadly known as grinding burn, a concept that includes different damages such as surface hardness variations, the development of tensile residual stresses, and, most harmfully, a drastic reduction in fatigue life that may lead to catastrophic failure in service. Although this problem is especially relevant in low-thermal-conductivity alloys [8], more common materials such as tool steels can also be seriously affected by grinding burn [9]. 

It becomes clear, therefore, that the prediction and measurement of workpiece temperature during grinding is a critical issue. A large number of analytical [10,11] and numerical models [12,13] for effective temperature prediction in grinding are available in the scientific literature, which shows the importance of the subject. Whatever prediction method is used, the validation of predictions requires effective methods for in-process temperature measurement. In a reference work, Xu and Malkin [14] compared existing temperature measuring techniques for grinding. The methods used included an embedded thermocouple and a two-color infrared detector and a foil/workpiece thermocouple.

Thermography is another method used for measuring grinding temperatures. However, the presence of grinding fluid all over the ground part avoids a correct measurement, so accurate grinding temperature measurements using thermography can only be obtained in cases of dry grinding. In spite of this, numerous works using this type of sensor can be found in the literature. Nadolny and Plichta [15] tried to measure the temperature during an internal cylindrical grinding process without cutting fluid. For this purpose, they made a slot in the cylindrical part occupying one-third of its perimeter in order to measure the temperature there by means of a thermographic camera. The conclusions of the work highlight the unreliability of the measurements obtained. Likewise, the fact that no cutting fluid was used means that the results are far removed from the industrial practice of the process. Brosse et al. [16] used a thermographic camera to measure the temperature field on the lateral surface of a part, obtaining satisfactory results but, once again, without the use of cutting fluid, a fundamental aspect in grinding.

In addition to thermographic cameras, pyrometers have also been used to measure temperatures on parts. The above-cited work by Xu and Malkin [14] presented results that showed good correlation with other measuring techniques. Liu et al. [17] used infrared radiation for the measurement of subsurface temperatures in a stainless steel creep-feed grinding process. Later, Urgoiti et al. [18] replicated this approach by using the information obtained from a pyrometer and relating temperatures with specific grinding energies. In this case, the authors made some holes in the bottom of the ground part controlling the distance of the hole to the ground surface. Then, they introduced optic fiber in order to measure the temperature in the bottom of the hole using a pyrometer. This way, the authors obtained reliable measurements of the sub-surface of the part. By combining experimental results and numerical modeling, the maximum temperatures on the ground surface could be obtained. Baumgart et al. [19] improved the system and adapted it for a cylindrical grinding process. It can be concluded that the difficulty of determining the exact position of the optical fiber inside a part in each pass introduces uncertainty in the measured temperatures. Also, the presence of a machined hole significantly modifies the temperature field, and this must be accounted for.

In addition to experimental methods for measuring in-process grinding temperatures, molecular dynamics simulations also serve as an important tool for temperature analysis in mechanical machining processes [20]. Zhou et al. [21] used molecular dynamics to simulate the material removal process during the abrasive machining of SiC substrates, revealing that the distribution and exposed height of abrasives influenced grinding temperatures and subsurface damage. Similarly, Zhao et al. [22] utilized molecular dynamics simulations to study grinding temperatures and residual stresses.

The main alternative to non-contact measurement methods are contact sensors, or more specifically, thermocouples. Embedded thermocouples have been widely used for temperature measurement in tribological science since the early days of research [23]. One can find multiple references to the use of these sensors in grinding starting from the 1990s [24]. Typically, the thermocouple is placed within a stationary frictional component, positioning its sensing hot junction a short distance away from the moving surface. Evidently, this kind of thermocouple does not provide a reading of the surface temperature itself but instead gives an indication of the temperature beneath the surface. Occasionally, inverse heat transfer techniques are employed to deduce the temperature and heat flow on the sliding surface using data obtained from these conventional thermocouples (for instance, [25]). For grinding operations, embedded thermocouples are an interesting alternative in creep-feed grinding [26], in which temperature gradients are considerably smaller than those in conventional grinding. Still, the problems related to sensor location with respect to the ground surface (which is continuously being removed), and the distortion of the temperature field produced by the machined hole, persist. In addition, the delay in the response of the thermocouple must be considered. Available calibration cards for commercial thermocouples have been produced in conditions of maximum temperature and gradients very different to those existing in grinding.

So-called “grindable thermocouples” have been extensively used for temperature measurement in grinding. The main advantage of these sensors is that the uncertainties of thermocouple placement inside the part are eliminated, and the impact of the measured signal delay is minimized. In the case of single-pole thermocouples, the friction components serve as the thermocouple’s electrodes, creating the measuring junction directly at the point of contact. For grinding, single-pole thermocouples have been proposed and tested, and results have been validated in different research, for instance, [27]. In [28], the authors focus their analysis on the grinding process of a part made of a heat-resistant material, with a complex profile and very low feed rates (feed rate 1.2 m/min). This means that the time during which the heat source is in contact with the thermocouple is about 0.25 s, which is high enough to allow a reliable response of the thermocouples. Unfortunately, in industrial shallow grinding operations, contact times are much shorter, which means it is challenging for this type of sensor to respond.

An alternative to single-pole thermocouples is provided by double-pole grindable thermocouples, which are composed of two parallel electrodes. The tips of the electrodes are in direct contact with the sliding surface, establishing the measuring junction through plastic deformations. The feasibility of using double-pole grindable thermocouples has been thoroughly revised in [29] for friction experiments under stationary and transient regimens. If compared with the grinding process, the sliding speed is similar, but the maximum temperatures are considerably lower (30 °C in the case of friction experiments; above 200 °C in the grinding of steels).

Double-pole thermocouples can be placed in an array configuration in order to simultaneously measure grinding temperatures at different points of the contact area between the wheel and workpiece. This novel and interesting approach was first presented in [30]. The authors used K-type Thin-Film Thermocouples (TFTCs) featuring a 0.15 mm thickness, and these were insulated with perfluoroalkoxy resin jackets. Additionally, during the experiments, precise control over the protrusion of the thermocouples was maintained, ensuring they remained within the 15–20 µm range. As the wheel traversed over the thermocouple array, the materials at the tips of the two poles came into contact, effectively closing the circuit. The grinding conditions were close to those in industrial applications (wheel speed of 26.9 m/s and workpiece speed of 10 m/min). The authors evaluated the response characteristics of the welded junction using a stable flame at 801 °C. The response time was approximately 0.015 ms.

The approach using array configuration was later used in [31] for the grinding of titanium. Throughout the grinding process, the interaction period between the grinding wheel and the specific measurement point on the workpiece surface was long (approximately 0.5 s) if compared with industrial shallow grinding operations. Simultaneously, the response time of the thermocouple employed in the experiment was approximately 300 milliseconds. Consequently, the thermocouple’s response time in recording the grinding temperature for this application aligned with the stipulated requirements. Other application fields of grindable thermocouples can be found in the grinding of gears [13,32]. In both cases, the maximum temperatures and thermal gradients are much lower than those found in conventional grinding.

The literature review shows that grindable thermocouples have been successfully used in creep-feed grinding operations, in which the table speed is low, and therefore, temperature gradients are not very steep. Under these conditions, the response time of grindable thermocouples is fast enough to produce accurate temperature measurements. However, in conventional grinding operations with faster table speeds, as most industrial operations are, the delay in the response of the thermocouple results in large errors in the maximum measured value. In this paper, the need for accurate calibration of the response of grindable thermocouples is studied as a prior step for signal integration to correct thermal inertia. This is especially important in conventional shallow grinding operations with high values of feed speed. In Section 2, the procedure for the integration of the raw signal obtained from grindable thermocouples is presented. Using a laser head and a pyrometer, a reference temperature signal was used to show that the maximum measured temperature by the grindable thermocouple is far from reality. Also, integration using a first-order system approach provided excellent results at a table speed of 10 m/min. Then, grinding experiments in industrial conditions were carried out, and their results are discussed in Section 4. The obtained values of temperature are compared with the widely accepted thermal model by Marinescu. The results show that, if the raw signal is directly used from the thermocouples, the deviation in the maximum temperature with respect to the theoretical model is 200 K. After integration using the calibration constants obtained for the ground junction, the error can be reduced below 15% even for feed speeds as high as 40 m/min. The main conclusion is that, following the described procedure, maximum grinding temperatures can be effectively measured using grindable thermocouples even at high values of table speed.

## 2. Correction of Thermal Inertia of Grindable Junctions

Temperature measurement through thermocouples relies on the thermoelectric effect, which involves the direct conversion of temperature disparities into electrical voltage. The concept of the “thermoelectric effect” encompasses three distinct phenomena: the Seebeck effect, the Peltier effect, and the Thomson effect. In line with this principle, when two dissimilar metal wires are connected at one end and left open at the opposite end, any temperature fluctuation at the junction between these wires results in the generation of a voltage differential at the open end of the circuit.

The reaction time of a K-type thermocouple is not immediate. Figure 1 represents the K-type thermocouple’s response to a sudden change in input. Notably, the response exhibits a certain delay when compared to the input signal. This delay is primarily attributed to thermal inertia. To accurately determine the temperature, it is essential for the thermocouple’s junction to be heated to the desired temperature. This necessity results in the time lag between the input and the obtained signal. The magnitude of this lag is influenced by various factors, including the thermocouple’s diameter, the quality of contact between the thermocouple and the object whose temperature is being measured, and the type of thermocouple used, amongst others.

The behavior observed in Figure 1 is governed by a first-order differential equation given by Equation (1).
(1)$$\tau \cdot \frac{dy}{dt}+y\left(t\right)=k\cdot F\left(t\right)$$
where *y*(*t*) is the temperature given by the thermocouple, *F*(*t*) is the real temperature to be measured, *τ* is the “time constant” or “response time”, defined as the time required to reach 63.2% of an instantaneous temperature change, and *k* is the maximum temperature measured by the thermocouple expressed as a percentage of the step value. *k* is usually close to 1 and it does not affect the lag observed in thermocouple measurement. For commercial contact thermocouples, the value of *τ* depends on the wire diameter. Thus, typical values range from *τ* = 0.007–0.015 s for 0.25 mm diameter to *τ* = 0.02–0.03 s for 0.50 mm diameter.

In the case of grindable thermocouples, the diameter of the junction cannot be accurately known a priori. Therefore, it was decided to calibrate the time constant once the junction was formed. Figure 2 shows the preparation of the ground junction in the workpiece that was ground. A slot of dimensions 0.65 mm × 0.44 mm was WEDM’ed for the location of the wires. After grinding, the junction was formed, and the thermocouple was ready for measurement. Then, the calibration of *τ* and *k* was carried out. To do so, the thermocouple hot junction was submerged in boiling water (step input of 100 °C). The test was repeated 10 times in order to ensure its repeatability. Average values of *τ* = 0.016 s (standard deviation 0.002) and *k* = 0.91 (standard deviation 0.054) were obtained. After calibration, it could be observed that the value of *k* was somewhat lower than that of commercially available thermocouples. In the case of grindable thermocouples, the junction was not as perfect as that of commercial thermocouples. This fact can explain a lower value of *k*. In any case, the integration of the differential equation resulted in a sound correction of this effect. 

The capability of the junction to accurately measure temperatures at feed velocities similar to those of industrial shallow grinding operations was tested using a laser head and a two-color pyrometer, as shown in Figure 3. This experiment simulated the moving heat source produced by the grinding wheel, with the advantage that the actual contact temperature could be effectively measured using the pyrometer.

The measurement area of the pyrometer was adjusted to the area of thermocouples. In order to verify the repeatability and accuracy of sensors disposal, six experiments were carried out, obtaining high repeatability in the measurements. In the experiment, the laser spot diameter was 2 mm in order to ensure that all the spots passed over the measurement zone (which was smaller). The feed speed was 10 m/min. This value of feed speed corresponded to the lower range of feed speed in conventional surface shallow grinding operations. The power of the laser head was set at 200 W. Figure 4 represents the temperature evolution during the test, as measured by the pyrometer (the reference, in blue); the actual measurement as collected by the thermocouple (orange plot); and the corrected temperature (gray plot) as obtained from the integration of the actual measurement using Equation (1). Integration was performed using the values *τ* and *k* obtained in the previous paragraph.

Some interesting conclusions can be drawn from the analysis of Figure 4. The first aspect to be noticed is that, even at low values of the feed speed, the junction was not able to follow the temperature gradient. The maximum temperature measured by the pyrometer was 1274 K, whilst the maximum temperature directly collected by the junction was only 1017 K. In other words, probably, for creep-feed operations at very low feed speeds, direct temperature measurement from the junction may be feasible. However, for shallow grinding operations in which the feed speed is typically well over 10 m/min, direct measurement from the junction cannot be used.

The integration of the direct measurement from the junction using Equation (1) and the previously calibrated values of *τ* and *k* were in excellent agreement with the reference in terms of maximum temperature and delay. Then, the maximum value of the gray curve was 1285 K, which was very close to the 1274 K value measured by the pyrometer. This result confirms the need for the effective integration of the direct signal from the junction to avoid the inertia of the thermocouple at feed speeds common in shallow grinding operations. Also, some noise could be observed in the integrated curve. This is a numerical effect due to the variable slope of the directly measured temperature, which is affected by noise. Since in grinding, the objective is the measurement of the maximum temperature in the contact wheel–workpiece, this effect can be neglected.

## 3. Materials and Methods

Once the need for signal integration at feed speeds typical of conventional grinding was stated, the sensor needed to be tested in industrial conditions. Therefore, grinding experiments and temperature measurements using grindable thermocouples were carried out.

Industrial grinding experiments were conducted on a three-axis CNC surface grinder, Blohm Orbit 36, from Blohm Jung GmbH manufacturer, Hamburg, Germany. The grinding wheel used for the experiments has a specification 4MBA46G12, manufactured by Abrasivos UNESA, Hernani, Spain. The coolant used was a 5% water-based oil emulsion. The experiments were conducted in a T-form workpiece in order to control the grinding width as well as the actual depth of cut. For each test, 10 grinding passes were carried out in order to ensure the repeatability and reliability of the results.

The specifications of the grinding wheel used in the experiments, together with the grinding conditions, are shown in Table 1. Five different grinding conditions were proposed, maintaining a nearly constant value of specific material removal rate (*Q_w_’*) (within the range of 8–10 mm^3^/mm·s) and varying the value of the grinding speed ratio (*q_s_*), from 40 to 300. Wheel speed (*v_s_*) was maintained as constant throughout the tests. The objective was to analyze the influence of the value of feed speed (*v_w_*) on the temperatures reached on the workpiece surface during a conventional (shallow) surface grinding process. Medium dressing conditions were used for the grinding tests (using a dressing overlap ratio of *U_d_* = 4.2 revolutions). AISI 1045 bearing steel hardened at 54 HRc was used as the work material. K-type grindable thermocouples were used together with an NI 783311-01 acquisition device from National Instruments, Austin, TX, USA, for temperature measurement. The power consumption was measured using a Model UPC-FR Universal Power Cell from Load Controls Inc., Sturbridge, MA, USA, together with a NI usb-6008 data acquisition card from National Instruments.

## 4. Results and Discussion

The ability of the ground thermocouple to record the maximum grinding temperature on the work surface was checked against the widely used thermal model proposed by Marinescu et al. [33]. The model is based on the analytical solution to the problem of conduction for a moving heat source on a semi-infinite body first proposed by Jaeger [34]. Based on that, the following compact equation (Equation (2)) for the estimation of the maximum temperature on the ground surface can be derived:(2)$$T_{max}=C\cdot \frac{q_{w}}{\beta_{w}}\cdot \sqrt{\frac{l_{c}}{v_{w}}}$$

Equation (2) represents the dependency of the maximum temperature on the heat flux to the workpiece (*q_w_*), the feed speed (*v_w_*), the actual contact length (*l_c_*), and the thermal diffusivity of the work material (*β_w_*). *C* is an integration constant that depends on the Peclet number, which in turn depends on the feed speed *v_w_*, the actual contact length between wheel and workpiece *l_c_*, and the thermal properties of the work material (thermal conductivity *k_w_*, density *ρ_w_*, and specific heat *c_w_*). From the Peclet number, the integration constant *C* can be obtained from Table 2.
(3)$$Pe=\frac{v_{w}\cdot l_{c}}{4\cdot \frac{k_{w}}{\rho_{w}\cdot c_{w}}}$$

The application of the model requires a hypothesis directly related to the grinding operation to be made. First, the actual contact length must be estimated. To do so, and following data from the literature [33], the contact length was considered to be between 1.5 and 3 times the geometrical contact length (*l_g_*). For this study, a value of *l_g_* = 2.25 × *l_c_* was selected. Second, the heat flux conducted towards the workpiece was obtained using Equation (4), where *R_w_* is the heat partition ratio to the workpiece. From the literature, using alumina wheels with vitreous bonding and good flushing [33], a reference value of *R_w_* = 0.75 was set.
(4)$$q_{w}=R_{w}\cdot \frac{P}{l_{c}\cdot b_{w}}$$

Finally, the thermal properties of the AISI 1045 bearing steel are *k_w_* = 34.3 W/m·K, *ρ_w_* = 7815 kg/m^3^, and *c_w_* = 506 J/kg·K. Using the complete set of data explained above, the input values for application of the model can be found in Table 3.

The ability of the grindable thermocouples to accurately measure the maximum grinding temperatures on the workpiece could then be analyzed. Figure 5 shows (blue dots) the maximum temperatures (average value) obtained from direct measurement from the grindable thermocouples as a function of *v_w_*. The deviation for each measurement is represented as well, and it ranges from 4% for a value of *v_w_* = 6 m/min to 10% for a *v_w_* = 40 m/min. Together with the measurements, the results from the analytical model are also represented (orange dots). As expected, both sets of data follow a similar trend, with the maximum temperature increasing with a reduction in feed speed *v_w_*. This effect is well known by experienced users of the grinding process; feed speed is increased to avoid grinding burns. However, a noticeable effect is observed in terms of deviation between the experimental and theoretical values. As feed speed is increased, the distance between direct measurement from the grindable thermocouple and the model also increases. Thus, the distance is minimum for a value of *v_w_* = 6 m/min (88 K between average values). As *v_w_* increases, the deviation also increases. For instance, for *v_w_* = 20 m/min, an error of 233 K is observed, with similar values for higher feed speeds. As a conclusion, it becomes obvious that grindable thermocouples cannot adequately follow the steep gradients existing on the workpiece surface. Therefore, the integration of the raw signal is required.

From the above results, the integration of raw measured data using Equation (1) was performed. Figure 6 shows the new experimental values of maximum temperature as obtained from integration (blue dots) and, again, the results from the analytical model. A comparison of the results confirms the validity of the approach. Although deviation is still more obvious at high values of *v_w_*, a clear improvement can be noticed. Now, the highest deviation corresponds to the feed speed of 40 m/min and can be quantified as 93 K. For the rest of the tests, deviations with respect to the theoretical model are in all cases below 20 K.

## 5. Conclusions

From the work carried out, the following conclusions have been drawn:

The literature shows that grindable thermocouples exhibit large potential for the measurement of grinding temperatures, which is a difficult task because of the limited access to the contact zone between the wheel and workpiece, and because of the extremely steep thermal gradients generated. Very scarce data on the application of grindable thermocouples at a table speed of over 10 m/min (typical in shallow grinding) can be found, with most of the applications being focused on creep-feed grinding operations.

In the case of grindable thermocouples, the diameter of the junction cannot be accurately known a priori. Therefore, the calibration of *τ* and *k* must be carried out once the junction is formed. Using this approach, average values of *τ* = 0.016 s (standard deviation 0.002) and *k* = 0.91 (standard deviation 0.054) were obtained, which are sound when compared with the values of commercial thermocouples. In order to adjust the calibration temperature to actual grinding temperatures, future works will examine the possibility of using high-temperature molten solders for calibration.

The response of the ground junction in terms of maximum temperature measurement was evaluated using a laser head and a pyrometer as the reference value. At a table speed of 10 m/min, the maximum temperature measured by the pyrometer was 1274 K, whilst the maximum temperature directly collected by the junction was 1017 K. After integration to avoid the effect of thermal inertia, the maximum value from the ground junction was 1285 K, which was very close to the 1274 K value measured by the pyrometer. This result confirms the need for the effective integration of the direct signal from the junction at feed speeds common in shallow grinding operations.

Grinding experiments were carried out to check the validity of the approach. Experimental results from ground joints were compared with the theoretical values of a widely used analytical model. When comparing the model with the raw data from the thermocouple, the deviation increased as the feed speed increased, with errors around of 200 K for feed speeds higher than 10 m/min. 

After integration using the calibration constants obtained for the ground junction, the highest deviation observed was reduced down to 93 K at *v_w_* = 40 m/min. For the rest of the tests, deviations with respect to the theoretical model were in all cases below 20 K.

The main conclusion is that, following the described procedure, maximum grinding temperatures can be effectively measured using grindable thermocouples even at high values of table speed.

## Figures and Tables

**Figure 1 sensors-24-01741-f001:**
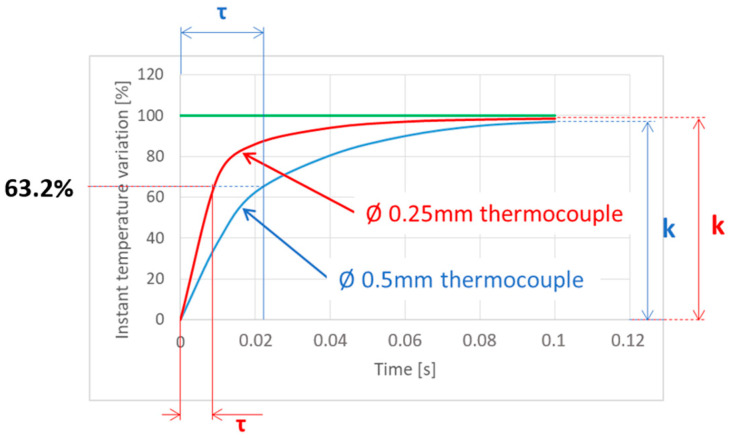
Response of contact thermocouple to a step input.

**Figure 2 sensors-24-01741-f002:**
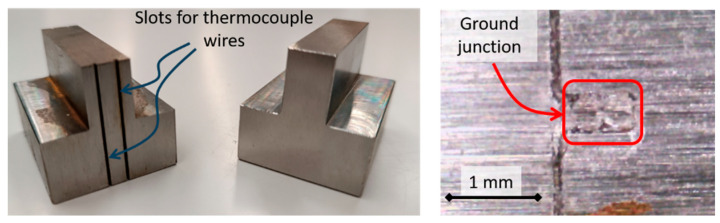
(**Left**): Part to be ground and WEDM’ed slots. (**Right**): Detail of the ground junction of the two wires of the thermocouple.

**Figure 3 sensors-24-01741-f003:**
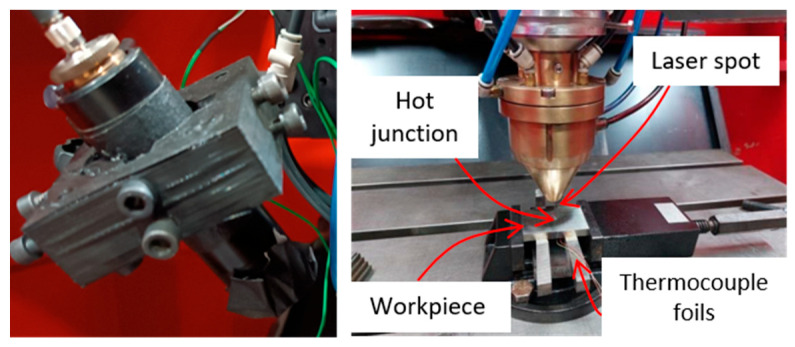
(**Left**): two color pyrometer. (**Right**): experimental set-up including laser spot, workpiece and foil thermocouples.

**Figure 4 sensors-24-01741-f004:**
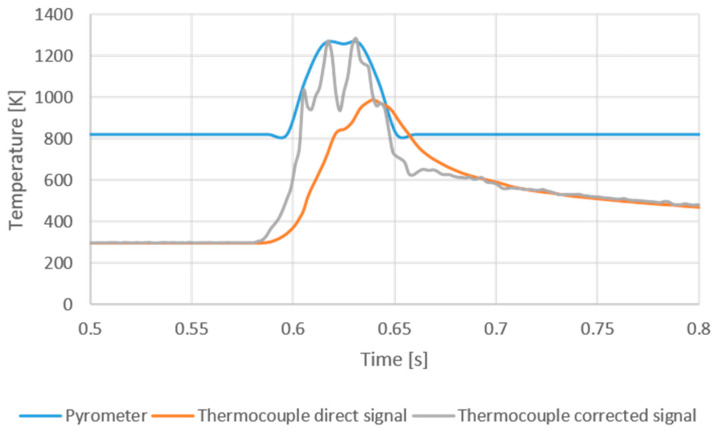
Temperature evolution during the experiment as measured by the pyrometer (reference), by the thermocouple, and the corrected signal obtained from Equation (1).

**Figure 5 sensors-24-01741-f005:**
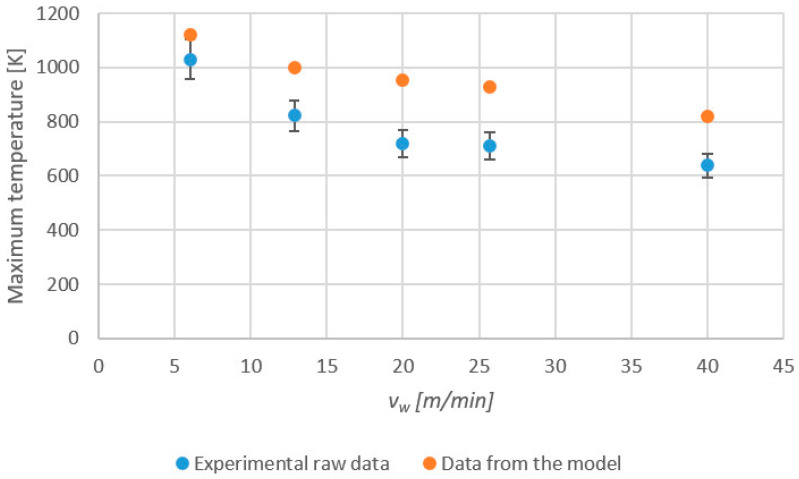
Maximum temperatures directly measured by the grindable thermocouples (blue dots) and estimated by the analytical model (orange dots) versus *v_w_*.

**Figure 6 sensors-24-01741-f006:**
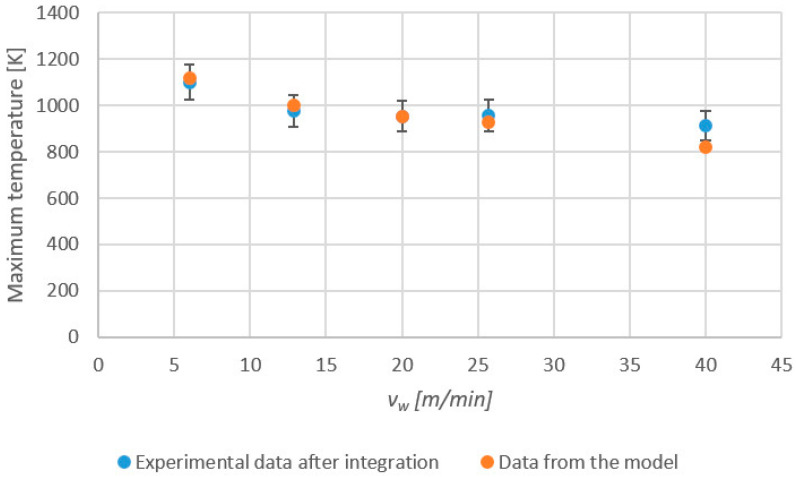
Maximum temperatures obtained from integration using Equation (1) (blue dots) and estimated by the analytical model (orange dots) versus *v_w_*.

**Table 1 sensors-24-01741-t001:** Industrial grinding experiments for temperature measurement.

Grinding Wheel		Work Material		Dressing Conditions	
Specification	4MBA46G12V	Specification	AISI 1045	*a_d_* [mm]	0.005
*d_s_* [mm]	383	Hardness	54 HRc	*v_fd_* [mm/min]	358
*b_s_* [mm]	40	*b_w_* [mm]	10	*v_s_* [m/s]	30
Grinding Tests					
Test	*v_w_* [m/min]	*q_s_* [-]	*a_e_* [mm]	*Q_w_’* [mm^3^/mm·s]	*Agg* [-]
1	40.0	45	0.012	8.31	127
2	25.7	70	0.019	8.16	101
3	20.0	90	0.025	8.49	91
4	12.9	140	0.044	9.38	76
5	6.0	300	0.104	10.42	55

**Table 2 sensors-24-01741-t002:** Integration constant *C* for different numbers of the Peclet number.

*Pe*	*C*
>10	1.06
0.2 < *Pe* < 10	$\frac{0.95}{\pi }\cdot \sqrt{2\cdot \pi +\frac{Pe}{2}}$
*Pe* < 0.2	0.76

**Table 3 sensors-24-01741-t003:** Set of input data for the thermal model.

Test	*l_g_* [mm]	*l_c_* [mm]	*Pe*	*C*	*q_w_* [W/mm^2^]
1	2.19	4.92	94.5	1.06	69.07
2	2.70	6.08	75.0	1.06	60.37
3	3.12	7.03	67.5	1.06	51.10
4	4.09	9.21	56.9	1.06	38.53
5	6.32	14.21	41.0	1.06	24.63

## Data Availability

The raw data supporting the conclusions of this article will be made available by the authors on request.

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
