# Peer review of "Accurate Measurement of Temperatures in Industrial Grinding Operations with Steep Gradients"

_sensors, 2024, doi:10.3390/s24061741_

Round 1

Reviewer 1 Report

Comments and Suggestions for Authors

I would like to appreciate the very nice work of the authors. The article is clear, well understandable, and shows the possibility of temperature measurement using grindable thermocouples. I have only a few comments/questions that do not influence the acceptance of the article. See attached pdf file.

Author Response

The authors gratefully acknowledge the corrections and annotations made by the reviewer, that find really interesting and have used to improve the quality of the article and will also use some of the ideas to conduct future research in order to improve or complete this investigation.

Please see all the complete response to the reviewer's corrections in the attachment.

Reviewer 2 Report

Comments and Suggestions for Authors

see file

Author Response

The authors gratefully acknowledge the corrections and annotations made by the reviewer, that find really interesting and have used to improve the quality of the article and will also use some of the ideas to conduct future research in order to improve or complete this investigation.

Please see the complete response to the reviwer's corrections in the attachment.

Reviewer 3 Report

Comments and Suggestions for Authors

This work investigated the accurate calibration of the response of the grindable thermocouples after integration using the calibration constants obtained for the ground junction. The results showed that the error could be reduced to 93 K even for feed speeds as high as 40 m/min and bellow 20 K for lower feed speeds. The results are interesting. However, from the reviewer's perspective, before accepted for publication, this study needs to be improved, and the following issues should be clarified.

1. Line 220, the authors stated that the experimental equipment included a two-color pyrometer, but the reviewer cannot find it in Fig. 3.

2. In Fig. 4, there is a distinct fluctuation in the corrected temperature compared to the data from Pyrometer, which will greatly affect the accuracy of temperature measurement. What is the reason for this phenomenon and can it be corrected through the algorithms?

3. In the grinding experiment, the power consumption was collected, but in the analysis, it was pointed out that the correlation between the power consumption and the research of this work is not significant. Therefore, this part can be removed from the manuscript.

4. In addition to accurate experimental measurements, molecular dynamics simulation is also an important means of analyzing the temperature during the mechanical machining process. The author can cite the following article (the fourth paragraph of the Introduction section) to enrich their introduction section.

[A] https://doi.org/10.1088/2631-7990/ad207f .

5. The authors have given the installation position of the thermocouples in Fig. 2, but the position is far from the actual grinding position and cannot collect real grinding temperature. The author's description of the grinding experimental plan in Fig. 5 is insufficient. Can a detailed description be provided?

6. The image quality of all the figures should be improved significantly to meet the publishing requirements of the Sensors.

Comments on the Quality of English Language

-

Author Response

The authors gratefully acknowledge the corrections and annotations made by the reviewer, that find really interesting and have used to improve the quality of the article and will also use some of the ideas to conduct future research in order to improve or complete this investigation.

Please find the complete response to the reviewer's corrections in the attachment.

Round 2

Reviewer 2 Report

Comments and Suggestions for Authors

The article may be published

Reviewer 3 Report

Comments and Suggestions for Authors

The revised manuscript can be accepted.